# Isoflurane Anesthesia’s Impact on Gene Expression Patterns of Rat Brains in an Ischemic Stroke Model

**DOI:** 10.3390/genes14071448

**Published:** 2023-07-14

**Authors:** Yana Y. Shpetko, Ivan B. Filippenkov, Alina E. Denisova, Vasily V. Stavchansky, Leonid V. Gubsky, Svetlana A. Limborska, Lyudmila V. Dergunova

**Affiliations:** 1Laboratory of Human Molecular Genetics, National Research Center “Kurchatov Institute”, Kurchatov Sq. 2, Moscow 123182, Russia; yana.sch2014@yandex.ru (Y.Y.S.); filippenkov-ib.img@yandex.ru (I.B.F.); bacbac@yandex.ru (V.V.S.); limbor.img@yandex.ru (S.A.L.); 2Department of Neurology, Neurosurgery and Medical Genetics, Pirogov Russian National Research Medical University, Ostrovitianov Str. 1, Moscow 117997, Russia; dalina543@gmail.com (A.E.D.); gubskii@mail.ru (L.V.G.); 3Federal Center for the Brain and Neurotechnologies, Federal Biomedical Agency, Ostrovitianov Str. 1, Building 10, Moscow 117997, Russia

**Keywords:** tMCAO, “long-term” anesthesia, “short-term” anesthesia, isoflurane, RNA-Seq, gene expression

## Abstract

Background: Ischemic stroke (IS) is one of the most severe brain diseases. Animal models with anesthesia are actively used to study stroke genomics and pathogenesis. However, the anesthesia-related gene expression patterns of ischemic rat brains remain poorly understood. In this study, we sought to elucidate the impact of isoflurane (ISO) anesthesia on the extent of ischemic brain damage and gene expression changes associated with stroke. Methods: We used the transient middle cerebral artery occlusion (tMCAO) model under long-term and short-term ISO anesthesia, magnetic resonance imaging (MRI), RNA sequencing, and bioinformatics. Results: We revealed that the volume of cerebral damage at 24 h after tMCAO was inversely proportional to the duration of ISO anesthesia. Then, we revealed hundreds of overlapping ischemia-related differentially expressed genes (DEGs) with a cutoff of >1.5; *Padj* < 0.05, and 694 and 1557 DEGs only under long-term and short-term anesthesia, respectively, using sham-operated controls. Concomitantly, unique DEGs identified under short-term anesthesia were mainly associated with neurosignaling systems, whereas unique DEGs identified under long-term anesthesia were predominantly related to the inflammatory response. Conclusions: We were able to determine the effects of the duration of anesthesia using isoflurane on the transcriptomes in the brains of rats at 24 h after tMCAO. Thus, specific genome responses may be useful in developing potential approaches to reduce damaged areas after cerebral ischemia and neuroprotection.

## 1. Introduction

Ischemic stroke (IS) is multifactorial disease conditioned by blood clot formation or embolus [1]. IS remains a significant problem within both developed and developing countries [2]. There is currently an increased prevalence of IS in the population, as well as its neurological consequences, such as neuronal death, loss of brain tissue, impaired neural circuits, and the resulting motor, sensory, and cognitive impairments [3,4,5]. This situation emphasizes the need for a deeper understanding of the neural changes that occur during stroke to develop effective interventions. Animal models are actively used for studying stroke genomics and pathogenesis. Impaired blood flow in the middle cerebral artery is considered the cause of a significant portion of all ischemic pathologies of the brain [6,7]. Reperfusion events that occur with IS in humans using thrombolysis techniques should also be reflected. Therefore, transient middle cerebral artery occlusion (tMCAO) is often used for stroke modeling. Undoubtedly, like any model, tMCAO has its limitations. For example, the thromboembolic clot model is more suitable for studying the pathophysiology of cardioembolic stroke in humans [8].

Recently, a number of local and genome-wide gene expression assays were used to detect transcriptomic response under tMCAO. In this way, genes and their networks involved in metabolic cell activity, stress responses, neurosignaling, inflammation, cell death and other processes were identified [9,10,11,12,13,14]. Notably, the tMCAO model uses anesthesia for experimental animals. The methods can differ based on the duration of anesthesia, which can also depend on the degree of ischemic brain damage. Isoflurane (ISO) is the most commonly used anesthetic and has undeniable advantages in the early onset of action and recovery due to its low blood solubility relative to other inhalational drugs. However, the nature of gene involvement and activity in the brain processes of ischemic rats under anesthesia of different duration remains largely unexplored. It was previously shown that ISO therapy may aid in the formation of new blood vessels after IR and further enhance the microvascular network of the central nervous system (CNS). ISO is able to reduce brain damage through the T-cell receptor signaling pathway and achieve cerebral protection by regulating the production of VEGF and CD34 via the Shh/Gli signaling pathway [15]. Some authors expanded the functional understanding of ISO action by studying the far-reaching effects of dysregulated miRNA/target genes on processes such as synaptic transmission, axon development, and cell apoptosis [16]. Using immunofluorescent staining and Western blotting, we detected that ISO post-conditioning contributes to anti-apoptotic effects after cerebral ischemia in rats through the ERK5/MEF2D signaling pathway [17] and may reduce IR injury by downregulating the expression of aquaporin 4 (AQP4), possibly related to the bone morphogenetic protein 4/Smad1/5/8 signaling pathway [18].

In this study, we sought to elucidate the impact of ISO anesthesia on the extent of ischemic brain damage and gene expression changes associated with stroke. We considered two variants of the tMCAO model in rats (time of occlusion 90 min) using ISO anesthesia (3% induction, 1.5% maintenance) under long-term and short-term conditions. Rats in the long-term group remained under anesthesia during the operation and the entire period of occlusion (90 min), while rats in the short-term group were under anesthesia during the operation to occlude the vessels and restore blood flow. A comparison of previously obtained magnetic resonance imaging (MRI) data revealed that volume of cerebral damage at 24 h after tMCAO was inversely proportional to the duration of ISO anesthesia. Then, using RNA-Seq, we identified hundreds of differentially expressed genes (DEGs) with a cutoff of >1.5 and *Padj* < 0.05 between long-term and short-term anesthesia conditions. Among them, overlapping DEGs reflected the common ischemia-related transcriptome responses of ipsilateral subcortical cells at 24 h after tMCAO. Simultaneously, we identified a number of non-overlapping genes that were unique for use under long-term and short-term anesthesia conditions. Such DEGs can record anesthesia-related gene expression patterns in the brains of ischemic stroke model rats. The enrichment analysis provided functional annotations of the gene sets. Thus, the contribution of the activity of cellular systems in the brain to anesthesia-related effects under ischemia was also evaluated.

## 2. Materials and Methods

### 2.1. Animals

Rats of the Wistar line (weight 200–250 g) were obtained from AlCondi Ltd., Moscow, Russia. Rats were male, white, and 2 months old. The animals were divided into sham operation (SO) and ischemia–reperfusion (IR) groups for transient middle cerebral artery occlusion (tMCAO) model implementation.

### 2.2. tMCAO Model in Rats

#### 2.2.1. tMCAO under Long-Term Isoflurane (ISO) Anesthesia

The tMCAO rat model was applied using the method of Koizumi et al. [19], as previously described [20]. Ischemic (IR-lt) and sham-operated (SO-lt) rats were subjected to long-term ISO anesthesia (3% ISO induction and 1.5% maintenance) during the operation and throughout occlusion (90 min), as previously described [20]. The tMCAO details are described in Appendix A.

#### 2.2.2. tMCAO Model in Rats under Short-Term ISO Anesthesia

The tMCAO rat model was also applied using the method of Koizumi et al. [19], as previously described [1]. Ischemic (IR-st) and sham-operated (SO-st) rats were subjected to short-term ISO anesthesia (3% ISO induction and 1.5% maintenance) only during vascular occlusion surgery and blood flow recovery, as previously described [1]. The tMCAO details are described in Appendix A.

#### 2.2.3. Magnetic Resonance Imaging (MRI)

MRI analysis of the characteristics and size of the ischemic injury in rat brains at 24 h after tMCAO, when the rats were alive, was carried out using small-animal 7 T ClinScan MRI (Bruker BioSpin, Billerica, MA, USA), as previously described [1]. During the MRI study, rats were anesthetized with ISO. Then, all IR-lt, IR-st, SO-lt, and SO-st rats were injected with chloral hydrate (300 mg/kg) and decapitated at 24 h after tMCAO. Ten animals were included in each experimental group.

### 2.3. RNA Isolation

The subcortical brain structures of the ipsilateral hemispheres of IR-lt, SO-lt IR-st, and SO-st rats were taken at a range of +2 to −2 mm from the bregma. The samples included the striatum. The cortical region was not included in the analysis. All samples were subjected to RNA isolation, as previously described [21]. The RNA integrity number (RIN) was at least 9.0, in accordance with capillary electrophoresis (Experion, BioRad, Hercules, CA, USA) testing.

### 2.4. cDNA Synthesis and Reverse-Transcription Polymerase Chain Reaction (RT–PCR) in Real Time

cDNA synthesis using oligo (dT)_18_ primers was conducted as previously described [21]. Then, RT–PCR was conducted using a QuantStudio™ 3 real-time PCR system (Thermo Fisher Scientific, Waltham, MA, USA), as previously described [21]. Primers specific to the genes studied were selected using an OligoAnalyzer Tool (Integrated DNA Technologies IDT, Coralville, IA, USA) and were synthesized by the Evrogen Joint Stock Company (Appendix A). The procedure for cDNA sample analysis was performed in triplicate.

### 2.5. Data Analysis and Statistics

#### 2.5.1. Real-Time RT-PCR Data

Each of the comparison groups (IR-lt, IR-st, SO-lt, and SO-st) consisted of five animals (n = 5) for real-time RT-PCR. The *Gapdh* reference gene was used to normalize the cDNA samples. Calculations were performed as previously described [22]. The values were calculated using the 2^−ΔΔCt^ method, and only differences with *p* < 0.05 were considered to be significant.

#### 2.5.2. RNA-Seq Data

The RNA-Seq data for IR-lt vs. SO-lt and IR-st vs. SO-st pairwise comparisons had previously been obtained using the Cuffdiff program in [20] and [1], respectively. Here, RNA-Seq data were subjected to further analysis. Three animals were included in each of the comparison groups (IR-lt, SO-lt IR-st, and SO-st). The cutoff for gene expression changes was >1.5-fold. Only differences with *p*-values adjusted using the Benjamini–Hochberg procedure (*Padj* < 0.05) were considered to be significant.

#### 2.5.3. Functional Annotation and Cluster Analysis

The DAVID v.2021 [23] and PANTHER [24] datasets were used to annotate the functions of the differentially expressed genes (DEGs). Only annotations with *Padj* < 0.05 were considered to be significant. Heatmapper (Wishart Research Group, University of Alberta, Ottawa, Canada) [25] was used to perform hierarchical cluster analysis of the DEGs. The regulatory networks were visualized using Cytoscape 3.8.2 software (Institute for Systems Biology, Seattle, WA, USA) [26], as previously described [22].

### 2.6. Availability of Data and Material

RNA-sequencing data were deposited in the SRA database under accession code PRJNA472446 (SAMN09235828-SAMN09235839, https://www.ncbi.nlm.nih.gov/Traces/study/?acc=SRP148632 (accessed on 6 February 2019)) [27], accession code PRJNA803984 (SAMN25694602-SAMN25694613, http://www.ncbi.nlm.nih.gov/bioproject/803984 (accessed on 18 April 2022)) [28].

## 3. Results

### 3.1. MRI Study of the Ischemic Injury of Rat Brains at 24 h after tMCAO under Long-Term and Short-Term ISO Anesthesia

MRI in the T2-weighted imaging (T2-WI) mode identified that 8 out of 10 rats in the IR-lt group had a focal lesion in the subcortex region of the right (ipsilateral) brain hemisphere at 24 h after tMCAO. Concomitantly, all 10 rats in the IR-st group had a focal hemispheric lesion in the subcortex region of the right (ipsilateral) brain hemisphere that had also spread to the adjacent cortex. A typical MRI of ischemic foci after tMCAO under long-term and short-term ISO anesthesia is shown in Appendix A, respectively. Stroke-score MRI data are also shown in Appendix A. Notably, the volume of ischemic foci under long-term ISO anesthesia was less than that under short-term ISO anesthesia at 24 h after tMCAO.

### 3.2. RNA-Seq Analysis of the Effect of IR on the mRNA Level in the Subcortical Structures of the IH Related to SO Rats under Long-Term and Short-Term Anesthesia at 24 h after tMCAO

In this study, we compared the expression profiles of 17,368 genes (mRNAs) in the subcortical brain structures of rats containing an ischemic lesion at 24 h after tMCAO under both long-term and short-term ISO anesthesia conditions. We previously identified 1939 and 2802 DEGs with cutoffs >1.5 fold and *Padj* < 0.05 using RNA-Seq and common bioinformatics in IR-lt vs. SO-lt [20] and IR-st vs. SO-st [1] pairwise comparisons, respectively. The results of data comparison for IR-lt vs. SO-lt and IR-st vs. SO-st are presented in a Venn diagram (Figure 1a). Concurrently, we identified 1245 common overlapping DEGs that altered expression relative to the corresponding brain controls of sham-operated (SO) rats. These common genes significantly changed expression (mainly enhanced their mRNA levels) in both variants of the tMCAO model. Specifically, between the IR-lt vs. SO-lt and IR-st vs. SO-st comparison groups we identified 799 overlapping DEGs that were codirectionally upregulated (Figure 1b) and 430 overlapping genes that were codirectionally downregulated (Figure 1c). Upregulated DEGs in both comparison groups included *Hspa1a*, *Ptx3*, *Atf3*, *Socs3*, and *Sdc1*, whereas the top five downregulated DEGs were *Acvr1c*, *Erbb4*, *Ltc4s*, *Grin2a*, and *Bhlhe22* (Figure 1d). Appendix A includes a full list of these DEGs.

Additionally, we found 694 DEGs with unique effects in long-term anesthesia-related transcriptomes (Figure 1a). These DEGs altered the expression levels at 24 h after tMCAO only under long-term anesthesia. Among them, 302 DEGs presented upregulated mRNA in IR-lt vs. SO-lt and 392 genes were downregulated in this pairwise comparison. Moreover, we observed that the top five upregulated DEGs were *S100a9*, *Slc5a3*, *Lyve1*, *Nudt15*, and *Rbm46*, which more than doubled the mRNA level. The top five downregulated DEGs in IR-lt vs. SO-lt were the *Dmkn*, *Ppp1r1b*, *Slc35d3*, *Egr4*, and *Shbg* genes (Figure 1e). A full list of DEGs unique to the IR-lt vs. SO-lt pairwise comparison is shown in Appendix A.

The Venn diagram in Figure 1a presents 1557 DEGs with unique effects in the IR-st vs. SO-st pairwise comparison associated with short-term anesthesia-related effects. Among them, 583 DEGs were upregulated in IR-st vs. SO-st. The top DEGs included genes encoding chemokines (*Ccl22*, *Ccl5*), proteins of the coagulation system (*F2rl1*), and others (Figure 1f and Appendix A). Simultaneously, 974 downregulated DEGs mainly encoded other components of the neurotransmission system (*Neurod2*, *Kcna5*, *Kcnq3*) (Figure 1f, Appendix A).

Figure 1g shows a hierarchical cluster analysis of all DEGs for the comparisons of IR-lt vs. SO-lt and IR-st vs. SO-st. The common DEGs between the comparison groups reflected the effects of IR in the rats under both conditions. Simultaneously, individual differences between the groups characterized the specific long-term and short-term anesthesia-related transcriptome responses in the IH.

### 3.3. Reverse-Transcription Polymerase Chain Reaction (RT-PCR) Validation of the RNA-Seq Results at 24 h after tMCAO under Long-Term and Short-Term Anesthesia

A real time RT-PCR analysis of the expression of four genes (*S100a9*, *Mmp9*, *Hes5* and *Klf4*) was used to validate the RNA-Seq results in the IR-lt vs. SO-lt and IR-st vs. IR-st pairwise comparisons. The *Mmp9* and *Hes5* genes were DEGs in both IR-lt vs. SO-lt and IR-st vs. SO-st. The *S100a9* gene was a DEG in IR-lt vs. SO-lt, but a non-DEG in IR-st vs. SO-st, whereas the *Klf4* gene was a DEG in IR-st vs. SO-st, but a non-DEG in IR-lt vs. SO-lt. The real-time RT-PCR method adequately confirmed the RNA-Seq results (Figure 2).

### 3.4. Divergent Changes in the Expression of Genes in Rat Brains under Long-Term and Short-Term Anesthesia 24 h after tMCAO

Comparison between only the upregulated DEGs in IR-lt vs. SO-lt and only the downregulated DEGs in IR-st vs. SO-st presented eight overlapped DEGs, as shown in Figure 3a. Additionally, eight overlapping DEGs, including both downregulated DEGs in IR-lt vs. SO-lt and upregulated DEGs in IR-st vs. SO-st, are illustrated in Figure 3b. In total, 16 DEGs demonstrated oppositely directed changes in expression. The gene expression profiles for the eight genes in Figure 3a (*Lrrc58*, *Ranbp2*, *Lifr*, *Gpr22*, *Sult1a1*, *Cldn1*, *Slc19a3*, *Zbtb41*) and eight genes in Figure 3b (*Apold1*, *Klf2*, *Egr2*, *Dusp1*, *Bcl6b*, *Slc4a11*, *Wfs1*, *Nr4a1*) are shown in Figure 3c. Using the PANTHER functional annotation software, we found that these genes encode proteins of transcriptional regulation, the functioning of ion channels, lipid metabolism, etc.

### 3.5. Functional Annotation of DEGs in the Subcortical Structures of Brain after tMCAO in Rats under Different Durations of Anesthesia

We compared functional annotations to identify overlapping and unique data between IR-lt vs. SO-lt and IR-st vs. SO-st pairwise comparisons based on the DAVID v.2021 annotation results. The Venn diagram presented in Figure 4a illustrates 111 overlapping signaling pathways. These pathways were annotated according to the most enriched KP annotations between IR-lt vs. SO-lt and IR-st vs. SO-st. The top five KP pathways among the overlapped annotations, the corresponding number of up- and downregulated DEGs, and the *Padj* values in IR-lt vs. SO-lt and IR-st vs. SO-st are shown in Figure 4b and Figure 4c, respectively. Mitogen-activated protein kinase (MAPK), proteoglycans in cancer, pertussis, and pathways in cancer were included among the top five pathways in accordance with their *Padj* values. IR predominantly activated the expression of genes related to these pathways. Concomitantly, IR also suppressed the expression of genes related to the neuronal system of oxytocin, calcium, glutamatergic synapse, and other pathways in IH, despite the different durations of anesthesia comparison groups.

Additionally, Figure 4a showed 14 and 43 KP annotations that lie outside the intersection and were unique in their effects of IR-lt and IR-st transcriptomes vs. corresponding SO controls. The data related to the most significant signaling pathways (with a minimal *Padj*) that were unique for IR-lt vs. SO-lt and annotated with KP are illustrated in Figure 4d, whereas similar data for IR-st vs. SO-st are illustrated in Figure 4e. We identified antigen processing and presentation, purine metabolism, the Jak-STAT signaling pathway, hematopoietic cell lineage, drug metabolism for other enzymes, and other pathways that were unique for IR-lt vs. SO-lt and predominantly associated with upregulated DEGs. Simultaneously, we found more powerful neuronal-related genetic responses specific for IR-st vs. SO-st—namely, nicotine addiction and morphine addiction were included in the top five pathways. We also observed neuroactive ligand–receptor interactions, serotonergic synapse axon regeneration, and other signaling pathways predominantly associated with downregulated DEGs in IR-st vs. SO-st. The KEGG enrichment results were mostly reproduced using RP and WP annotations (Appendix A).

### 3.6. Functional Annotation of DEGs That Reversed Their Expression Levels in Rat Brains after tMCAO under Different Durations of Anesthesia

Next, we analyzed 16 DEGs (*Apold1*, *Klf2*, *Egr2*, *Dusp1*, *Bcl6b*, *Slc4a11*, *Wfs1*, *Nr4a1*, *Lrrc58*, *Ranbp2*, *Lifr*, *Gpr22*, *Sult1a1*, *Cldn1*, *Slc19a3*, and *Zbtb41*) that changed their expression under long-term and short-term ISO anesthesia conditions in opposite directions. These DEGs were annotated with the KP, RP, and WP pathways using DAVID v. 2021.

As a result, there were 43 KP, 23 RP, and 6 WP annotations for 11 out of 16 DEGs. Five genes (*Apold1*, *Bcl6b*, *Slc4a11*, *Lrrc58*, *Zbtb41*) had no pathway associations. Then, we chose annotations that had significant (*Padj*) associations with the IR-lt vs. SO-lt or IR-st vs. SO-st gene sets. Ultimately, nine genes (*Apold1*, *Bcl6b*, *Slc4a11*, *Lrrc58*, *Ranbp2*, *Gpr22*, *Sult1a1*, *Slc19a3*, *Zbtb41*) had no significant pathway associations, whereas 21 KP, 2 RP, and 2 WP annotations, as well as 7 DEGs (*Nr4a1*, *Klf2*, *Cldn1*, *Egr2*, *Lifr*, *Dusp1*, *Wfs1*) associated with these pathways formed the nodes of a functional network (Figure 5). The pathways formed two clusters. Cluster 1 united 16 pathways that overlapped between the IR-lt vs. SO-lt and IR-st vs. SO-st pairwise comparisons. Cluster 2 included six short-term anesthesia-related pathways that were unique for IR-st vs. SO-st. Notably, the *Nr4a1*, *Egr2*, and *Lifr* genes were involved in the maximal number of pathways (five) in the network (Figure 5). These genes also had the maximal number of intracluster connections. Thus, each of the *Nr4a1* and *Egr2* genes was involved in four pathways of cluster 1, whereas the *Lifr* gene was associated with three pathways of cluster 2. Interestingly, all genes, except for the *Klf2* and *Wfs1* genes, were associated with pathways from both clusters (Figure 5).

Notably, the signaling pathways of both clusters were predominantly associated with an immune response. Most genes that had any connections in the network were upregulated for IR-st vs. SO-st, but downregulated for IR-lt vs. SO-lt (Figure 5). Nevertheless, the network demonstrates heterogeneity between the immune responses under different durations of anesthesia at 24 h after tMCAO. Thus, the pathways of cell adhesion molecules, cytokine–cytokine receptor interaction, leukocyte transendothelial migration, PI3K-Akt, and other signaling pathways were associated with common effects of ischemia (cluster 1), whereas the immune system, MAPK, JAK-STAT, and others were short-term anesthesia-related signaling pathways (cluster 2) (Figure 5).

## 4. Discussion

In the present study, we identified the impact of ISO anesthesia on the extent of ischemic brain damage and gene expression changes associated with stroke. Two variants of the tMCAO model with different durations of ISO anesthesia were implemented. Using MRI, we observed that the long-term volume of ischemic foci was less than that under short-term ISO anesthesia at 24 h after tMCAO. Notably, most rats in the IR-lt group presented focal lesion in the subcortex region of the right (ipsilateral) brain hemisphere at 24 h after tMCAO. Concomitantly, all rats in the IR-st group had a focal hemispheric lesion in the subcortex region of the right (ipsilateral) brain hemisphere, which spread to the adjacent cortex. Numerous studies have demonstrated that ISO has a neuroprotective effect characterized by various mechanisms associated with the activation of γ-aminobutyric acid and glycine receptors, ionic channel antagonism, and alterations to the activities and functions of other cellular proteins [29]. It was proven that ISO, at clinically relevant concentrations, provides overall brain protection during hypoxia or ischemia by inhibiting cortical electrical activity and cerebral metabolism [30,31,32,33]. Despite its minimal metabolism, ISO can produce neurotoxic metabolites [34]. Single ISO exposure causes less extensive damage than multiple exposures, which can cause long-term cognitive dysfunction and apoptotic cell death in the developing brain [35]. Previous studies showed that ISO-induced anesthesia increases an inflammation factor (TNF-α) and oxidative stress marker (4HNE), which additionally contribute synergistically to neurotoxicity [36].

Then, using two variants of the tMCAO model under long-term and short-term anesthesia, we identified genes that had mRNA expression profiles both dependent on and independent of the duration of anesthesia at 24 h after tMCAO. As a result, four groups of genes in the subcortical structure of the rat brain were identified.

The first group included more than 1000 genes that overlapped between the pairwise comparisons and codirectionally changed the expression levels in both variants of the tMCAO model. Additionally, these genes reflected the overall effects of ischemia in brain cells in an anesthesia-independent manner. Among them, these genes were those encoding heat-shock proteins, components of cytokine signaling, apoptosis, and inflammatory and immune responses. The activities of these processes have been observed under ischemia in numerous studies [20,37,38,39]. Most genes from the first group increased their mRNA levels during ischemia using both model variants. The results could reflect the activation of processes associated with massive damage and help overcome the consequences of this damage via the immune system. The RNA-Seq data for some genes in the first group were verified by our team using PCR techniques on extended animal brain samples after tMCAO. The first group of genes included *Mmp9* and *Hes5*, which were up- and downregulated, respectively, under both conditions of anesthesia. The matrix metalloproteinase 9 (MMP-9) protein of the *Mmp9* gene was shown to be activated during inflammation and can be used as a prognostic indicator of stroke severity [40]. Previously, under long-term anesthesia based on our Western blot analysis of the IR rat subcortex, we observed an approximately 1.7-fold increase in total content of the MMP-9 protein compared to the SO rat group [41]. This effect is expected to be similar under short-term anesthesia. Transcription factor HES-5 of the *Hes5* gene, which is activated downstream of the Notch signaling pathway, plays a significant role in cell differentiation regulation in several tissue types [42]. It was shown that the *Hes5* gene is downregulated alongside the blockage of Notch signaling, which subsequently leads to a reduction in Hes5-STAT3 complex formation and the hypophosphorylation of STAT3, which attenuates the expression of manganese superoxide dismutase (MnSOD) and enhances reactive oxygen species (ROS) and apoptosis. In particular, *Hes5* gene overexpression protects hepatocytes from apoptosis after IR through the activation of STAT3 and MnSOD expression [43,44].

The second group of genes included 694 DEGs whose expression levels changed significantly under long-term, but not short-term, anesthesia at 24 h after tMCAO. Notably, the patterns of transcriptional activity among these genes largely reproduce the immune and inflammatory responses of ischemia in the rat brains. Functional clustering revealed the activity of antigen presentation and processing, JAK-STAT, and other systems. Based on the MRI data, rats receiving prolonged ISO anesthesia predominantly retained greater cell viability in the right hemisphere. Moreover, some studies have suggested that the use of anesthesia is an effective method to reduce stroke damage after IR and improve neurological functioning [45]. It is likely that the processes that ensure the regeneration of damaged cells, the utilization of dead cells, and the processes of compensation for functions lost by the surrounding tissue can occur more intensively. Furthermore, it was demonstrated that the infiltration of peripheral immune cells occurs in areas remote from the sites of primary injury during the progression of injury and brain repair [46,47]. Thus, the reaction of immune-related genes may be reasonable. Based on PCR, we revealed *S100a9* upregulated genes in extended animal brain samples after tMCAO under long-term but not short-term anesthesia. Some studies indicate that calcium-binding S100A8 and S100A9 proteins are highly represented under inflammatory conditions and expressed in neutrophil and monocyte cytosol [48]. Moreover, the S100A8 and S100A9 complex plays a decisive role in the control of macrophage-mediated kidney recovery after IR injury [49]. In this process, neutrophils are released from the bone marrow, and determine the direction of migration to the inflammation site in response to inflammation against lipopolysaccharides (LPS) [50]. Similar effects of *S100a9* upregulation based on the progression and quality of postischemic recovery can be expected under neuroprotective long-term ISO anesthesia conditions.

The third group of genes was the most numerous: more than 1500. The mRNA levels for these genes changed significantly under short-term but not long-term anesthesia at 24 h after tMCAO. The expression profiles of these genes reflect the effects associated with enlarged damaged area in the brain. Indeed, the MRI results of ischemic rats under short-term anesthesia indicated increased focus that extended to the adjacent cortex. Interestingly, the PCR results confirmed the RNA-Seq expression results for the *Klf4* gene, which was upregulated only under short-term anesthesia at 24 h after tMCAO. Previously, anti-inflammation Krüppel-like factor 4 (KLF4) encoded by the *Klf4* gene was found to relieve cerebral vascular injuries by improving vascular endothelial inflammation and regulating the expression of tight-junction proteins after ischemic stroke [51]. In addition, KLF4 was among the factors of pluripotency in vivo whose expression caused a number of neuroprotective effects [52]. In a recent study, Wang and Li suggested the critical role of KLF4 in regulating the activation of A1/A2 reactive astrocytes following ischemic stroke. Thus, the patterns of this key gene, according to our data, are associated with the duration of anesthesia and the degree of brain tissue damage [53].

Most genes from the third group significantly reproduced the neurotransmitter-related effects of ischemia. Downregulated genes were primarily clustered in functional categories associated with neuroactive ligand–receptor interactions and synapse activity. It is known that damaging processes in stroke affect neuroreceptor systems through excitotoxicity and synaptic plasticity impairment [54,55,56]. At the same time, at the transcriptome level, we showed that the process of reducing neurotransmitter-related gene expression (receptors, neurosignaling molecules and their processing) can be reversible. Thus, when using the neuroprotective peptide drug Semax (a synthetic analogue of adrenocorticotropic hormone), we demonstrated the activation of neurotransmitter genes during ischemia [57]. This observation may indicate that the use of long-term anesthesia with a neuroprotective component would have a similar effect. This is likely why these neurotransmitter-related genes altered their expression under short-term but not long-term anesthesia at 24 h after tMCAO. As a practical conclusion, we suggest that researchers take into account the background effects of anesthesia on the transcriptome when testing the transcriptome activity of neuroprotective drugs. Our results could help decrease such background effects.

The fourth group of genes is also interesting. Out of the overlapping comparisons, we identified 16 genes that exhibited opposite changes in expression levels under both variants of the tMCAO model. Interestingly, these genes were divided equally into two subgroups. There were eight genes (*Apold1*, *Klf2*, *Egr2*, *Dusp1*, *Bcl6b*, *Slc4a11*, *Wfs1*, *Nr4a1*) that increased their expression levels under short-term anesthesia but decreased their expression levels under long-term anesthesia. Eight other genes (*Lrrc58*, *Ranbp2*, *Lifr*, *Gpr22*, *Sult1a1*, *Cldn1*, *Slc19a3*, *Zbtb41*) were instead downregulated under short-term anesthesia but upregulated under long-term anesthesia in the subcortical structures of rat brains at 24 h after tMCAO. Using functional annotation tools, we discovered major genes association with immune responses. Moreover, significant annotations were found for genes of the first subgroup. These annotations formed two clusters of signaling pathways. Cluster 1 united pathways that could be involved in both short-term and long-term anesthesia-related processes in ischemic brains. Cluster 2 included pathways that were unique for short-term anesthesia-related events. Many studies have reported that genes of the fourth group play an important role in ischemia. Thus, the *Nr4a1* gene was discovered to be a transcription factor that plays an important role in the regulation of rapid responses to inflammatory modulators and other irritants after ischemia. Krüppel-like factor 2 (KLF2) encoded by the *Klf2* gene protects BV2 microglial cells against oxygen and glucose deprivation injury by modulating the BDNF/TrkB pathway [58]. Additionally, a mutation in the *WFS1* gene (c.1756G>A p.A586T) is responsible for early clinical features of cognitive impairment and recurrent ischemic stroke [59]. The gene expression profile identified in this study may indicate specific roles of these genes as integrators or switches in the regulation of tissue damage and regeneration after ischemia and protective action, including ISO anesthesia.

Ultimately, the second, third, and fourth groups of genes described mainly reflected the influence of ISO anesthesia duration in rat brain cells at 24 h after tMCAO. Based on the MRI data obtained, the ischemic focus under long-term anesthesia is smaller than under short-term anesthesia. Thus, the area containing viable cells grows larger as the duration of ISO anesthesia increases. Thus, the DEGs identified in these three groups could be associated with processes located at different distances from the stroke focus. In our study, we used RNA-sequencing and RT-PCR technologies, but did not analyze noncoding RNA or proteomic regulatory levels. It would also be useful to compare the effects of different anesthetics (isoflurane, sevoflurane, desflurane, methoxyflurane) at different doses of administration under ischemic conditions. However, the limitations of the present study could be overcome by expanding our research approaches in the future.

## 5. Conclusions

In our study, we elucidated the impact of ISO anesthesia on the extent of ischemic brain damage and gene expression changes associated with stroke using two variants of the tMCAO rat model under both long-term and short-term anesthesia. As a result, we revealed that the volume of cerebral damage at 24 h after tMCAO was inversely proportional to the duration of ISO anesthesia. Concomitantly, unique DEGs identified under short-term anesthesia were mainly associated with neurosignaling systems, whereas unique DEGs identified under long-term anesthesia were predominantly related to inflammatory response. Thus, specific genome responses may be useful in developing potential approaches to reduce damaged areas after cerebral ischemia and neuroprotection. These findings could be also applied in biotechnology and biomedicine as the genetic basis for developing new anesthetic techniques and testing systems.

## Figures and Tables

**Figure 1 genes-14-01448-f001:**
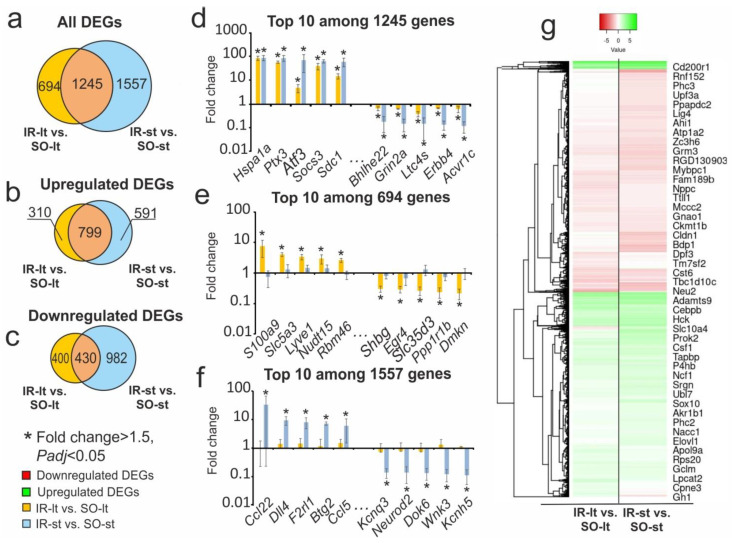
The RNA-Seq analysis of gene expression changes recorded 24 h after both tMCAO models under long-term and short-term ISO anesthesia. (**a**–**c**) Comparisons of the results obtained in the two pairwise comparisons of IR-lt vs. SO-lt and IR-st vs. SO-st in subcortical structures of the IH are represented using a Venn diagram. All (**a**), upregulated (**b**), and downregulated (**c**) DEGs are shown for comparison. (**d**–**f**) The top ten genes among 1245 (**d**), 694 (**e**) and 1557 (**f**) featuring the greatest fold changes in pairwise comparisons are also presented in a Venn diagram (**a**). The data are presented as the mean ± standard error (SE) of the mean. (**g**) Hierarchical cluster analysis of all DEGs in IR-lt vs. SO-lt and IR-st vs. SO-st, where each row represents a DEG; n = 3 per group.

**Figure 2 genes-14-01448-f002:**
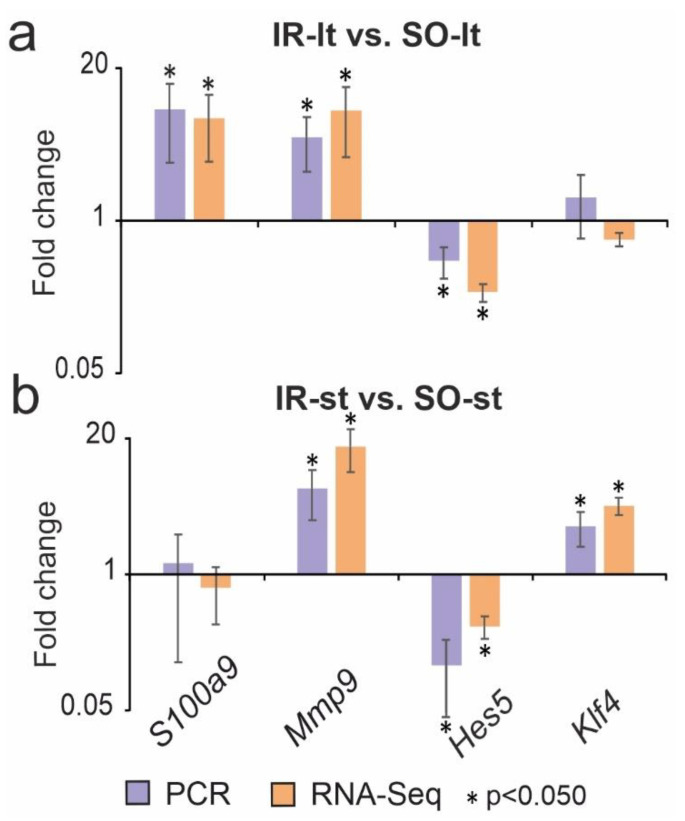
Real-time RT-PCR validation of the RNA-Seq results. Data for comparisons of IR-lt vs. SO-lt and IR-st vs. SO-st are presented in (**a**) and (**b**) respectively. The reference gene *Gapdh* was used to normalize the expression of the cDNA samples; n = 5 per group.

**Figure 3 genes-14-01448-f003:**
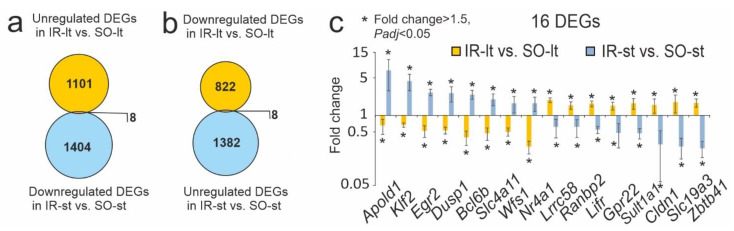
A comparison of the RNA-Seq outcomes revealed that focal IR induced multidirectional changes in gene expression in rat brains under long-term and short-term anesthesia 24 h after tMCAO. (**a**,**b**) Venn diagrams presenting overlaps between the IR-lt vs. SO-lt and IR-st vs. SO-st DEGs. We illustrate comparisons between IR-lt vs. SO-lt upregulated DEGs and IR-st vs. SO-st downregulated DEGs (**a**) and vice versa between IR-lt vs. SO-lt downregulated DEGs and IR-st vs. SO-st upregulated DEGs (**b**). (**c**) In total, 16 DEGs were present within the intersection of the gene sets on the Venn diagram (**b**). These genes were downregulated for IR-lt vs. SO-lt, but upregulated for IR-st vs. SO-st. The data are presented as the mean ± SE of the mean.

**Figure 4 genes-14-01448-f004:**
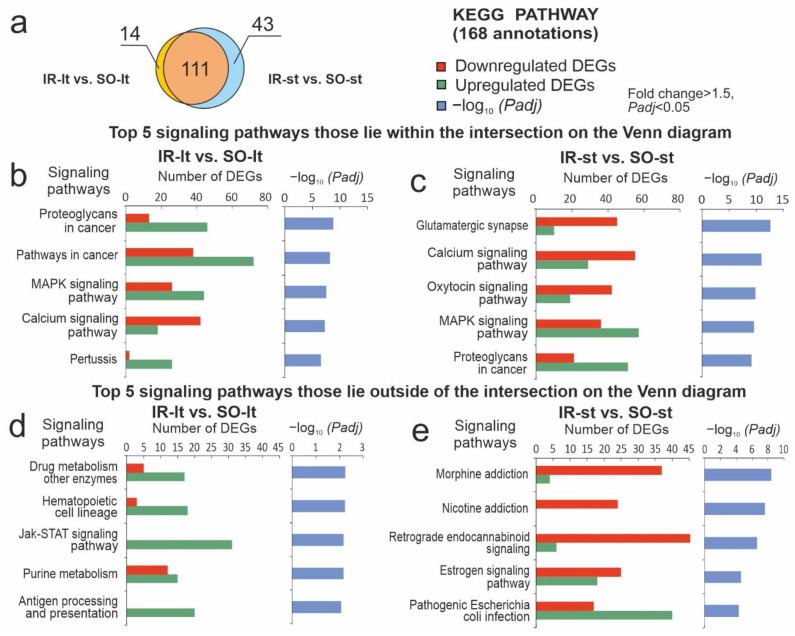
Ischemia and anesthesia-related signaling pathways associated with DEGs at 24 h after tMCAO. DAVID v. 2021 was used to enrich DEGs via KEGG PATHWAY (KP) annotations (pathways). (**a**) A Venn diagram presenting the results of a comparison between IR-lt vs. SO-lt and IR-st vs. SO-st KP annotations. The number of annotations is indicated in the diagram segments. (**b**,**c**) Top five significant pathways that lay within the intersection on the Venn diagram (**a**). Each pathway was included based on the *Padj* values, as well as the number of corresponding up- and downregulated DEGs in IR-lt vs. SO-lt (**b**) and IR-st vs. SO-st (**c**). (**d**,**e**) Top five significant pathways that lay outside of the intersection on the Venn diagrams (**a**). Each pathway was included based on the *Padj* values, as well as the number of corresponding up- and downregulated DEGs in IR-lt vs. SO-lt and (**d**) in IR-st vs. SO-st (**e**). Only DEGs and KP annotations with *Padj* < 0.05 were considered to be significant; n = 3 per group.

**Figure 5 genes-14-01448-f005:**
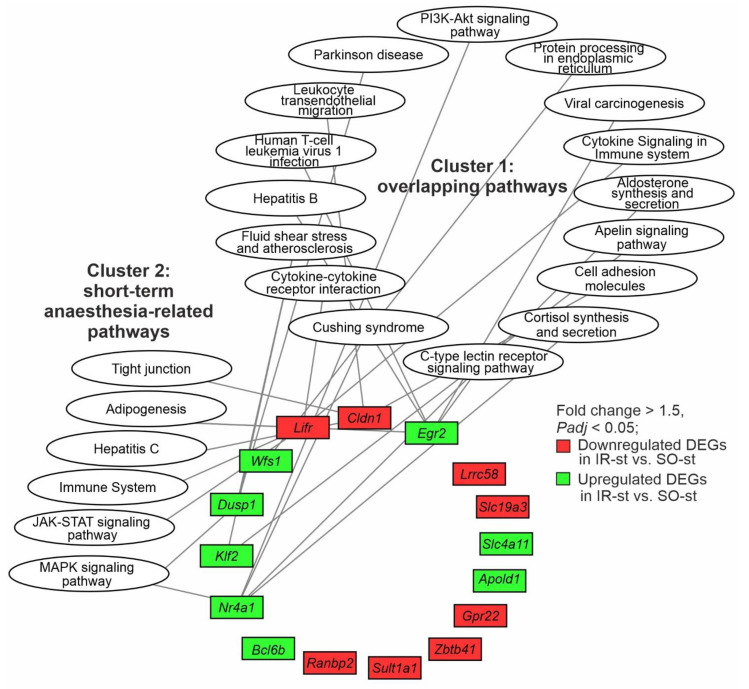
The DEGs with reversed mRNA levels in rat brains at 24 h after tMCAO under long-term and short-term anesthesia and gene-associated pathway annotations. Cytoscape 3.8.2 software was used for network construction. Involvement of the protein product of the gene in signaling pathway functioning is indicated by a line connecting the nodes. The KP, RP, and WP databases were used to annotate all clustered signaling pathways. Only DEGs and annotations with *Padj* < 0.05 were considered to be significant.

## Data Availability

Publicly available datasets were analyzed in this study. These data can be found here: [27,28].

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
