# Peer review of "Isoflurane Anesthesia’s Impact on Gene Expression Patterns of Rat Brains in an Ischemic Stroke Model"

_genes, 2023, doi:10.3390/genes14071448_

Round 1
Reviewer 1 Report
Genes (genes-2476841)
Isoflurane Anesthesia Duration Influence on the Brain Transcriptome in Ischemic Stroke Rat Model
This manuscript presents a study of the impact of isoflurane anesthesia duration on gene expression in a rat model of ischemic stroke (tMCAO).
The authors use a rat model in which the MCA is transiently occluded. The authors either used isoflurane for the entire occlusion time (90 min, “long-term”) or just for the procedures to occlude and restore flow (“short-term”). At least five rats were used in each group. Rats were imaged with MRI at 24 hours and then euthanized with RNA extracted from the “subcortical structures”. RNA expression was evaluated by RNA-Seq as well as RT-PCR for selected genes. Gene expression results are further evaluated into functional groups by DAVID and PANTHER approaches.
The authors found that longer treatment with isoflurane, during the occlusion, resulted in smaller stroke volumes. The authors also compared gene expression in tMCAO with long-term, short-term isoflurane to sham occlusion with long-term or short-term isoflurane.
This is a valuable study on an important topic. The topic is introduced adequately. Methods need to be improved, in part by moving some critical information from the supplementary figures. The results are presented clearly. The interpretation and discussion of the results would benefit from considering the issues listed below.
Significant issues:
1. Significant effort is needed to improve grammar. The manuscript is difficult to read in its current state.
2. Please include exact number of rats in each group, not just “at least 5”.
3. It is unclear what subcortical brain structures were used for analysis. The description of “+2 to -2 mm from bregma” needs to be clarified.
4. Was anesthesia used for MRI imaging and euthanasia? The supplementary figure is clear but the information should be in the Methods.
5. The discussion would benefit more efforts to put the differences between the groups into some meaningful structure. This is understandably difficult with hundreds of genes, but perhaps more effort to focus on existing stroke literature would be helpful. Many of the referenced gene roles seem removed from the current study.
Minor issues:
1. Line 168 needs a space between “ofrats”.
2. Parts of supplementary tables are not in English (e.g. worksheet names).
The English grammar needs work, and while much of the grammar is technically correct the text overall is disjointed and difficult to follow.
Author Response
Response to the comments of Reviewer 1 to Manuscript ID: genes-2476841
Authors:
We are very grateful to the Reviewer 1 for the review and constructive comments. We carefully considered the comments of the Reviewer 1 and attached the answers to all comments.
Reviewer 1:
- Significant effort is needed to improve grammar. The manuscript is difficult to read in its current state.
Authors:
In accordance with the Reviewer’s recommendation, changes were added in the text. The manuscript was undergone by English correction by MDPI English editing services: Please see order details [Manuscript ID: English-68416] MDPI English editing - Editing completed and English-Editing-Certificate-68416 in attachment.
Reviewer 1:
- Please include exact number of rats in each group, not just “at least 5”.
Authors:
In accordance with the Reviewer’s recommendation, changes were added in the text (lines 148, 149, 174, 304 in Mark-up copy).
Reviewer 1:
- It is unclear what subcortical brain structures were used for analysis. The description of “+2 to -2 mm from bregma” needs to be clarified.
Authors:
In accordance with the Reviewer’s recommendation, changes were added in the text (lines 151-153 in Mark-up copy). The subcortical brain structures of the ipsilateral hemispheres of IR-lt, SO-lt IR-st, and SO-st rats were taken at a range of +2 to -2 mm from the bregma. The samples included the striatum. The cortical region was not included in the analysis.
Reviewer 1:
- Was anesthesia used for MRI imaging and euthanasia? The supplementary figure is clear but the information should be in the Methods.
Authors:
In accordance with the Reviewer’s recommendation, changes were added in the text (lines 146-149 in Mark-up copy). During the MRI study, rats were anesthetized with ISO. Then, all IR-lt, IR-st, SO-lt, and SO-st rats were injected with chloral hydrate (300 mg/kg) and decapitated at 24 h after tMCAO.
Reviewer 1:
- The discussion would benefit more efforts to put the differences between the groups into some meaningful structure. This is understandably difficult with hundreds of genes, but perhaps more effort to focus on existing stroke literature would be helpful. Many of the referenced gene roles seem removed from the current study.
Authors:
We are grateful to the Reviewer 1 for the comments, but the discussion of the role of several genes in accordance with the current study matter was substantially provided in the text (e.g. lines 470-486, 501-512, 517-528, 547-572 in Mark-up copy). In accordance with the Reviewer’s recommendation, the graphical abstract was added to put the differences between the groups into more meaningful structure.
Reviewer 1:
Minor issues:
- Line 168 needs a space between “ofrats”.
Authors:
In accordance with the Reviewer’s recommendation, changes were made in the text (line 216 in Mark-up copy).
Reviewer 1:
- Parts of supplementary tables are not in English (e.g. worksheet names).
Authors:
In accordance with the Reviewer’s recommendation, changes were made in Supplementary Tables.

Reviewer 2 Report
Shpetko and colleagues in the present article entitled ‘Isoflurane Anesthesia Duration Influence on the Brain Transcriptome in Ischemic Stroke Rat Model’, explore the effects of isoflurane anesthesia on gene expression in a transient middle cerebral artery occlusion (tMCAO) model of ischemic stroke. Here, the authors aimed to elucidate the impact of isoflurane anesthesia on the extent of ischemic brain damage and gene expression changes associated with stroke. The study utilized the tMCAO model to induce ischemic stroke in animal subjects, with isoflurane anesthesia administered during the procedure.
In general, I think the idea of this article is really interesting and the authors’ fascinating observations on this timely topic may be of interest to the readers of Biomedicines. However, some comments, as well as some crucial evidence that should be included to support the author’s argumentation, needed to be addressed to improve the quality of the manuscript, its adequacy, and its readability prior to the publication in the present form. My overall judgment is to publish this paper after the authors have carefully considered my suggestions below, in particular reshaping parts of the ‘Introduction’ and ‘Methods’ sections by adding more evidence.
Please consider the following comments:
• I suggest changing the title. In my opinion, in the present form it is too wordy and it seems to be not enough clear and specific.
• Abstract: In my opinion, Authors should consider rephrasing this section. According to the Journal’s guidelines, the Abstract should contain most of the following kinds of information in brief form. Please, consider giving a more synthetic overview of the paper's key points: I would suggest rephrasing the results and conclusion to make them clear for readers to understand.
• A graphical abstract that will visually summarize the main findings of the manuscript is highly recommended.
• Introduction: The introduction lacks clarity and organization. It would benefit from a clear and concise overview of the research objectives, hypotheses, and the significance of the study. Additionally, I would suggest to add some background information on the prevalence and impact of ischemic stroke, as well as emphasize the neurological consequences of ischemic stroke, such as neuronal death, loss of brain tissue, impaired neural circuits, and the resulting motor, sensory, and cognitive impairments ((https://doi.org/10.3389/fpsyg.2022.1044988; DOI: 10.3390/biomedicines11030945; https://doi.org/10.3389/fpsyt.2023.1225755). This information would help in explaining the need for a deeper understanding of the neural changes that occur during stroke to develop effective interventions.
• Transient Middle Cerebral Artery Occlusion (tMCAO) Model in Rats: The description of the transient middle cerebral artery occlusion (tMCAO) model is brief and lacks important details. The Authors should provide more information on the rationale for using this model, the specific procedures involved, and any limitations or potential confounding factors associated with the model.
• Results: The paper does not adequately address potential confounding factors that may influence the results. For example, the duration of anesthesia and its potential impact on the degree of ischemic brain damage are briefly mentioned but not thoroughly explored. The authors should discuss how variations in anesthesia duration might have affected the outcomes and how they controlled for such confounding factors. Furthermore, the paper mentions the identification of differentially expressed genes (DEGs) using RNA-Seq, but it does not provide detailed results or discuss the functional implications of these findings. The authors should include more comprehensive data analysis, such as fold changes and statistical significance, and provide a more in-depth interpretation of the identified DEGs in relation to ischemic stroke and anesthesia.
• Finally, the paper lacks a thorough discussion of the limitations of the study and potential directions for future research. It would be beneficial to address any methodological limitations, potential biases, and areas for improvement in the experimental design. Additionally, the authors should highlight possible future studies to build upon their findings and address any remaining gaps in knowledge.
• In my opinion, I think the ‘Conclusions’ paragraph would benefit from some thoughtful as well as in-depth considerations by the authors, because as it stands, it is very descriptive but not enough theoretical as a discussion should be. Authors should make an effort, trying to explain the theoretical implication as well as the translational application of their research.
• References: Authors should consider revising the bibliography, as there are several incorrect citations. Indeed, according to the Journal’s guidelines, they should provide the abbreviated journal name in italics, the year of publication in bold, the volume number in italics for all the references.
I hope that, after these careful revisions, this paper can meet the Journal’s high standards for publication.
I am available for a new round of revision of this article.
I declare no conflict of interest regarding this manuscript.
Best regards,
Reviewer
Minor editing of English language required.
Author Response
Response to the comments of Reviewer 2 to Manuscript ID: genes-2476841
Authors:
We are very grateful to the Reviewer 2 for the review and constructive comments. We carefully considered the comments of the Reviewer 2 and attached the answers to all comments.
Reviewer 2:
- I suggest changing the title. In my opinion, in the present form it is too wordy and it seems to be not enough clear and specific.
Authors:
In accordance with the Reviewer’s recommendation, the Title was changed (lines 1-4 in Mark-up copy).
Reviewer 2:
- Abstract: In my opinion, Authors should consider rephrasing this section. According to the Journal’s guidelines, the Abstract should contain most of the following kinds of information in brief form. Please, consider giving a more synthetic overview of the paper's key points: I would suggest rephrasing the results and conclusion to make them clear for readers to understand.
Authors:
In accordance with the Reviewer’s recommendation, changes were added in Abstract section (lines 18-43 in Mark-up copy).
Reviewer 2:
- A graphical abstract that will visually summarize the main findings of the manuscript is highly recommended.
Authors:
In accordance with the Reviewer’s recommendation, the graphical abstract was added to visually summarize the main findings of the manuscript.
Reviewer 2:
- Introduction: The introduction lacks clarity and organization. It would benefit from a clear and concise overview of the research objectives, hypotheses, and the significance of the study. Additionally, I would suggest to add some background information on the prevalence and impact of ischemic stroke, as well as emphasize the neurological consequences of ischemic stroke, such as neuronal death, loss of brain tissue, impaired neural circuits, and the resulting motor, sensory, and cognitive impairments ((https://doi.org/10.3389/fpsyg.2022.1044988; DOI: 10.3390/biomedicines11030945; https://doi.org/10.3389/fpsyt.2023.1225755). This information would help in explaining the need for a deeper understanding of the neural changes that occur during stroke to develop effective interventions.
Authors:
In accordance with the Reviewer’s recommendation, changes were added in the Introduction section (lines 50-54, 65-68, 73-88, 103, 104 in Mark-up copy).
Reviewer 2:
- Transient Middle Cerebral Artery Occlusion (tMCAO) Model in Rats: The description of the transient middle cerebral artery occlusion (tMCAO) model is brief and lacks important details. The Authors should provide more information on the rationale for using this model, the specific procedures involved, and any limitations or potential confounding factors associated with the model.
Authors:
In accordance with the Reviewer’s recommendation, details about tMCAO model were added (lines 55-63 in Mark-up copy). Also, technical information regarding the tMCAO model procedure under long-term and short-term anesthesia was described in Supplementary Method S1 and S2, respectively.
Reviewer 2:
- Results: The paper does not adequately address potential confounding factors that may influence the results. For example, the duration of anesthesia and its potential impact on the degree of ischemic brain damage are briefly mentioned but not thoroughly explored. The authors should discuss how variations in anesthesia duration might have affected the outcomes and how they controlled for such confounding factors. Furthermore, the paper mentions the identification of differentially expressed genes (DEGs) using RNA-Seq, but it does not provide detailed results or discuss the functional implications of these findings. The authors should include more comprehensive data analysis, such as fold changes and statistical significance, and provide a more in-depth interpretation of the identified DEGs in relation to ischemic stroke and anesthesia.
Authors:
In accordance with the Reviewer’s recommendation, the potential impact on the degree of ischemic brain damage was discussed (lines 431-451 in Mark-up copy). The potential confounding factors were controlled by the following experimental design: only white 2-month-old male rats (weight, 200–250 g) were used; time tracking of anesthesia duration was conducted (Supplementary Figure S1); MRI was used to control the volume of ischemic damage; RT-PCR was used to validate RNA-Seq results, as described in Materials and Methods section. The functional annotation of RNA-Seq results was presented in Supplementary Table S6, as well as in the Description column of Supplementary Tables S3-S5. Also, comprehensive data analysis, such as fold changes and statistical significance was provided in the manuscript (lines 32, 113, 191, 191, 232, 233 in Mark-up copy). Additionally, statistical criteria were provided at the end of Supplementary Tables S3-S6. We believe that our findings could be also applied in biotechnology and biomedicine as the genetic basis for developing new anesthetic techniques and testing systems (lines 600-601 in Mark-up copy).
Reviewer 2:
- Finally, the paper lacks a thorough discussion of the limitations of the study and potential directions for future research. It would be beneficial to address any methodological limitations, potential biases, and areas for improvement in the experimental design. Additionally, the authors should highlight possible future studies to build upon their findings and address any remaining gaps in knowledge.
Authors:
We are grateful to the Reviewer 2 for the comments, but the limitations the study and potential directions for future research, including studies of the non-coding RNA and proteomic regulatory levels was provided in the Discussion section (lines 579-581, 583-585 in Mark-up copy). In accordance with the Reviewer’s recommendation, the Conclusion section was additionally expanded (lines 581-583 in Mark-up copy).
Reviewer 2:
- In my opinion, I think the ‘Conclusions’ paragraph would benefit from some thoughtful as well as in-depth considerations by the authors, because as it stands, it is very descriptive but not enough theoretical as a discussion should be. Authors should make an effort, trying to explain the theoretical implication as well as the translational application of their research.
Authors:
In accordance with the Reviewer’s recommendation, changes were added in the Conclusion section (lines 587-598 in Mark-up copy).
Reviewer 2:
- References: Authors should consider revising the bibliography, as there are several incorrect citations. Indeed, according to the Journal’s guidelines, they should provide the abbreviated journal name in italics, the year of publication in bold, the volume number in italics for all the references.
Authors:
We are grateful to the Reviewer 2 for the comments. In accordance with the Reviewer’s recommendation, the revising of bibliography was made and changes were added. It should be noted that Mendeley Desktop reference manager with pre-upload citation style of the journal Genes was used.

Round 2
Reviewer 2 Report
The authors did an excellent job clarifying all the questions I have raised in my previous round of review. Currently, this paper is a well-written, timely piece of research and provides a useful description of neural mechanisms underlying ischemic stroke and its neurological consequences, exploring the role of gene expression in the pathophysiology of ischemic stroke and examines its impact on neural damage and recovery.
Overall, this is a timely and needed work. It is well researched and nicely written, with a good balance between descriptive and narrative text.
I believe that this paper does not need a further revision, therefore the manuscript meets the Journal’s high standards for publication.
I am always available for other reviews of such interesting and important articles.
Reviewer